# Imaging Plant Lipids with Fluorescent Reporters

**DOI:** 10.3390/plants14010015

**Published:** 2024-12-25

**Authors:** Yong-Kang Li, Guang-Yi Dai, Yu-Meng Zhang, Nan Yao

**Affiliations:** 1Guangdong Provincial Key Laboratory of Plant Stress Biology, State Key Laboratory of Biocontrol, School of Life Sciences, Sun Yat-sen University, Guangzhou 510275, China; liyk27@mail2.sysu.edu.cn (Y.-K.L.); zhangym87@mail2.sysu.edu.cn (Y.-M.Z.); 2South China National Botanical Garden, Chinese Academy of Sciences, Guangzhou 510275, China; daigy@foxmail.com

**Keywords:** plant lipids, visualization, fluorescence, biosensor, spatiotemporal dynamics

## Abstract

In plants, lipids function as structural elements and signaling molecules. Understanding lipid composition and dynamics is essential for unraveling their biological functions and metabolism. Mapping the spatiotemporal distribution of lipids in plants holds great potential for elucidating lipid biosynthetic pathways and gaining insights to guide crop genetic engineering. Recent progress in fluorescence microscopy and imaging has opened new opportunities for researchers to visualize plant lipids *in vivo* at high spatiotemporal resolution. In this review, we provide an up-to-date overview of the methods used to image plant lipids with fluorescence microscopy. We highlight caveats and potential limitations of these approaches and provide suggestions for optimizing their utilization. This review synthesizes current knowledge and highlights the potential of these methods to provide new insights into lipid biology.

## 1. Introduction

Lipid molecules exhibit substantial structural and functional diversity and play crucial roles in essential activities within living organisms, including serving as structural elements and signaling molecules [1,2,3]. For example, cellular membranes are primarily composed of lipids, which influence the fluidity and curvature of the membrane, regulate membrane protein activity, and recruit peripheral proteins [4].

The lipid composition of membranes varies among organelles, even those of the same type in the same cell; in addition, lipids can be asymmetrically distributed between the two leaflets of the membrane bilayer and even within the same leaflet. These variations result from interactions between lipids and proteins, metabolic pathways, lipid transporters, and vesicular transport pathways [5,6,7]. Membranes can also contain a mix of lipid physical states, including gel/solid and liquid disordered or ordered phases [8,9]; these states depend on the lipid composition, packing affinity, and environmental conditions. Lipid charge also affects membrane electrostatics, which contribute to organelle identity and protein localization [10]. Moreover, lipids present in lesser amounts can have outsized effects on the membrane electrostatic field. That is to say, small differences in lipid structure and abundance can have profound effects on crucial biological functions. Therefore, considering the diverse structures and distributions of lipids is important for studying cells of different tissues or organisms.

Researchers have been able to accurately and quantitatively detect lipids in cellular fractions for almost 25 years. For example, fluorophore-based enzymatic assays are sensitive methods for detecting lipids [11]. Nuclear magnetic resonance is another reliable means for analyzing the lipids in living plants and seeds [12,13], and mass spectrometry is widely used for the quantitative detection of plant lipids. Coherent anti-Stokes Raman scattering (CARS) microscopy is a non-linear vibrational imaging technique that allows for label-free visualization of cellular lipid storage by harnessing the natural vibrational signatures of lipid molecules, providing high-contrast images of lipid droplets within cells [14]. Although these methods can measure lipids in bulk, visualizing lipids provides additional information that is crucial for understanding lipid functions.

Visualizing lipids using microscopy and specific fluorescent dyes or tags can help confirm quantitative results and reveal the spatiotemporal dynamics of plant lipids. To facilitate analysis of lipids via microscopy, researchers have generated or identified various dyes and lipid analogs containing functional fluorescent groups. These tools can be used for co-localization studies by combining lipid probes with different spectral proper-ties or in combination with fluorescent proteins. This approach is powerful, although the intrinsic fluorescence of some rare lipids, such as cholestatrienol and dehydroergosterol, can confound the results [15]. Moreover, the fluorescent dyes in current usage can detect molecules with the same structural property but cannot differentiate among specific lipid molecules or chemical structures. For example, Nile Red binds lipids based on its ability to interact with hydrophobic structures. To address this problem, researchers have developed genetically encoded fluorescent biosensors that can bind specific lipid molecules at physiologically relevant spatial and temporal scales *in vivo*. Although each method has its limitations and inherent weaknesses, together, they have made significant contributions to recent studies.

In this review, we provide a comprehensive overview of the approaches used to visualize plant lipids using fluorescence microscopy, including descriptions of the different types of available dyes, lipid analogs, and genetically encoded biosensors. By providing a thorough overview of the available tools and techniques, including their limitations and applications, we aim to guide researchers in selecting the most appropriate methods for their studies.

## 2. Dissecting Lipid Status with Fluorescent Dyes

Fluorescent dyes consist of a core structure (i.e., aromatic ring skeleton) that defines the dye’s absorption/emission wavelengths and core structure-modifying elements that alter its binding properties (Figure 1). The photophysical properties of these dyes may change according to the surrounding environment or specific molecular interactions. For example, the optical properties of the commonly used dye Laurdan depend on the polarity of the environment the dye molecules are immersed in, due to changes in their dipole moment upon excitation. The use of fluorescence-based reporters has greatly contributed to the analysis of lipid localization and distribution in plants (Table 1); in the next paragraphs, we give examples of these contributions.

Sterols are isoprenoid derivatives with a characterized planar sterol backbone made up of four condensed aliphatic rings and a hydrocarbon side chain at C17. Higher plants possess a vast array of sterols referred to as phytosterols. Sterols and sphingolipids form lipid clusters in the plasma membranes, providing a medium for protein signaling complexes and serving as key regions of membrane contact during infection [53,54]. Filipin labeling, which can be performed on aldehyde-fixed samples, specifically detects sterol complexes in plants. This method largely preserves fluorescent proteins and is compatible with immunocytochemistry [32]. Importantly, it can also be used to probe live cells, although it inhibits sterol-dependent endocytosis [33]. Successful protocols for single and combined dye staining based on several fluorescent lipid analogs, including FM4-64, Lissamine rhodamine B-phosphoethanolamine (LRB-PE), DiIC12, DiIC18, DiD, BODIPY-labeled C12 sphingomyelin (BD-SM), and the spectrally sensitive dye Laurdan, have been developed for staining sphingolipid-enriched and non-sphingolipid-enriched regions in the plasma membrane of live Arabidopsis (*Arabidopsis thaliana*) protoplasts [17]. Fluorescence recovery after photo-bleaching (FRAP) was employed to measure lipid dynamics, and time-dependent lipid polarization events, where the distribution of lipids, particularly sterols, within the cell is dynamically regulated over time to achieve cellular polarity, were observed to verify the viability of these staining protocols [18]. The trafficking mechanisms by which sterols move through the plant and into target cells remained unknown until the internalization of sterols was shown to primarily occur through a non-endocytic pathway in elongating Arabidopsis root cells through the use of fluorescent dehydroergosterol (DHE), BODIPY-cholesterol (BCh), FM4-64, filipin, and Nile Red labeling [55].

Fluorescent dyes can be used to monitor certain physiological behaviors of the membrane system, such as lipid trafficking and membrane organization. For instance, Di-4-ANEPPDHQ and the amphiphilic styryl dye FM1-43 were successfully used to monitor endo- and exocytosis in *Lilium longiflorum*, *Agapanthus umbellatus* L’Her, and tobacco (*Nicotiana tabacum*) cv. Samsun pollen tubes, and the same processes were monitored in Arabidopsis seedlings using FM4-64 [16,19,20,21,22,23]. Other markers, such as Di-4-ANEPPDHQ, DPH, TMA-DPH, and Laurdan, have also been used to monitor membrane organization through steady-state fluorescence spectroscopy. This analysis has helped uncover the roles of phytosterols and *Xanthomonas campestris* pv. campestris outer membrane vesicles in the organization and lateral heterogeneity of the plant plasma membrane [24,56]. Furthermore, the combination of Di-4-ANEPPDHQ and confocal laser scanning microscopy can be used to visualize membrane microdomains in living plant cells due to the phase separation characteristics of this marker [25,26,57].

Lipid-rich organelles play crucial roles in various plant physiological processes; for instance, lipid droplets (LDs) have dynamic functions in transient storage of neutral lipids, membrane remodeling, lipid signaling, and stress responses [58,59]. Nile Red exhibits vibrant fluorescence in a neutral lipid environment. Staining LDs with Nile Red allowed researchers to examine the effect of nutrient depletion on triacylglycerol accumulation in *Chlamydomonas reinhardtii* [60]. To demonstrate the effectiveness of spatially resolved Raman microspectroscopy in determining the iodine values (reflecting the degree of lipid unsaturation) in lipid storage bodies of individual living algal cells, a Nile Red fluorescence image of LDs within algal cells was compared to a Raman scattering image of the same living cell [27]. In another fluorescence study of LDs, three thalidomide analogs were synthesized and identified as markers for LDs in plants. These analogs emit blue light, making them easily combinable with green and red fluorescent reporters for multicolor live-cell imaging [29]. The co-localization of Nile Red–stained LDs with the PtOLE6-eGFP fusion protein (PtOLE6 is an LD-specific OLEOSIN protein, and eGFP is enhanced green fluorescent protein [GFP]) was confirmed in all cell types of transformed Arabidopsis [61]. This confirmation facilitates the study of LD motility and the potential trafficking routes through the cytosol to target plasmodesmata. Chloroplast outer envelope membranes were visualized in several plant species, including crops, using live-cell staining with the fluorescent dyes rhodamine B and Nile Red [28]. Both GFP-AtSDP1 and mCHERRY-AtPEX11e have a close association or are engulfed by LDs, as shown by BODIPY or Nile Red staining; this information was used as an example to introduce this method for protein localization analysis in plants [35].

These methods have proven to be effective for detecting lipids involved in various physiological processes such as cell polarization, ion transport, and phagocytosis, and visualizing their subcellular localizations within living cells. However, these dyes are currently unable to differentiate among specific lipid molecules; therefore, additional studies using techniques such as mass spectrometry and other lipid measurements are needed to draw definitive conclusions about lipid composition. In fluorescence-based lipid imaging, it is imperative to recognize that the efficacy of different reporters can vary significantly when employed for identical experimental objectives. This variation stems from the disparate mechanisms underlying the function of each fluorescent dye. For example, it was revealed that frequently used polarity-sensitive probes, Laurdan and Di-4-ANEPPDHQ, probe different properties of the lipid membrane [62]. As Amaro et al. (2017) pointed out, since di-4-ANEPPDHQ is an electrochromic dye, the electrochromic property of di-4-ANEPPDHQ can substantially bias the interpretation of empirical values gathered from cell biology experiments. Therefore, it is crucial to discern the mechanisms and relative merits of reporters by referring to specific research papers or by conducting comparative experiments when choosing a reporter for plant lipid studies.

## 3. Probing Lipids in Plant Using Fluorescent Tags

The development of lipid analogs containing fluorophores has greatly advanced the visualization of lipid distribution and metabolism. Lipid analogs have slight chemical modifications that do not alter their important functional groups (Figure 2). However, attaching a fluorescent tag to a lipid molecule to create a lipid analog can potentially affect the lipid’s function due to several factors, including changes in molecular size, polarity, and interactions with other cellular components [63]. These molecules can be used to probe the lipid distribution and lipid metabolism in cells by various methods (Table 1). For example, to assess the impact of aluminum on the activity of the phosphatidylcholine (PC)-hydrolyzing enzyme phospholipase C, labeled lipid BODIPY-PC can be used to produce labeled lipid products within cells [36].

Establishing and maintaining proper cell polarity are crucial for plant growth, as cell polarity affects cellular morphogenesis, reproduction, and the response to pathogen invasion. Polar cell growth relies on coordinated changes in the cell membrane and cell wall in response to the cellular environment. To confirm PtdIns(4,5)P_2_ delivery, pollen germination medium was supplemented with BODIPY tetramethyl-rhodamine-XC6-PtdIns(4,5)P_2_ (BODIPY-PIP2), and the taken images showed that both wild-type and *pip5k4* pollen tubes incorporate the fluorescently labeled PtdIns(4,5)P_2_ [20]. In another study of lipids in tip growth, tobacco (*Nicotiana tabacum*) cell line BY-2 (Bright Yellow-2) and pollen tubes were labeled with a BODIPY derivative of phosphatidylcholine (BODIPY-PC) to analyze pollen tube growth [37]. Another study used BODIPY-PC and BODIPY-DAG to examine the impact of aluminum on the expression, activity, and function of the nonspecific phospholipase C4 (NPC4) [36]. Carboxy-(5-(and-6)-carboxy-2′,7′-dichlorofluorescein diacetate) (carboxy-DCFDA) was used to label vacuoles in the pollen tubes of lilies (*Lilium formosanum* and *L. longiflorum*) and tobacco (*N. tabacum* cv. Samsun), and the pollen tubes were then imaged under a confocal microscope [21,37]. Additionally, BCh and DHE were used to label a region of native sterol accumulation, specifically the tips of emerging root hairs [55].

Lipid analogs are incorporated into plant cells through mechanisms similar to those of natural lipids, which might include endocytosis, membrane fusion, or direct transmembrane transport. Therefore, lipid analogs are valuable tools for exploring lipid behavior, from model membranes to functional membranes. However, the distribution of lipid analogs may not be identical to that of natural lipids due to the interaction of the fluorescent tags with cellular mechanisms that recognize and transport lipids to specific areas within the cell. To achieve selective affinity between fluorophore-labeled lipids and membrane assemblies, fluorescent lipid analogs with various fluorophores at the headgroup position and various lengths of the polyethylene glycol (PEG) spacer between the lipid backbone and fluorophore were investigated [64]. Unfortunately, most commercially available fluorescent lipid analogs suited for real-time imaging were designed to stain mammalian rather than plant cells. For example, nitrobenzoxadiazole (NBD)-labeled lipids, which are popular fluorescent membrane probes for use in artificial lipid bilayers and in animal and yeast plasma membranes, have not been optimized for use in plant cells [65,66,67]. To date, these NBD-labeled lipids, such as NBD-PC and NBD-sphingosylphosphocholine, were primarily used to visualize their own distribution and endogenously produced metabolic products in soybean (*Glycine max*) root cells [38], as well as in protein–lipid overlay assays *in vitro* [68].

Nevertheless, there have been recent developments in this area, with ongoing efforts to produce more engineered lipids and optimize experimental protocols. For instance, NBD-PC and 1-NBD-dodecanoyl-2-hydroxy-sn-glycero-3-phosphocholine (NBD-lyso-PC) were used to confirm the internalization of lyso-PC and glycerophospholipids by AtA-LA10 in planta by observing the labeled phospholipids inside the epidermal cells of the root tip following incubation [22]. The analysis of NBD-lipid uptake demonstrated that the ALA10–ALIS1 complex can internalize sphingolipids in Arabidopsis [69]. Although only a few fluorophores are currently available that can label lipid molecules without altering their physicochemical properties or biological functions, we anticipate that the identification and tracking of lipid movement in cellular systems will be significantly enhanced with the development of fluorescently labeled lipids that possess selective affinity for heterogeneous membranes in plants.

## 4. Genetically Encoded Biosensors Provide New Opportunities and Challenges

Genetically encoded biosensors based on fluorescent proteins enable researchers to noninvasively observe the inner workings of cells and organisms. By fusing fluorescent proteins to a sensing module, these biosensors can detect ligands in target tissues or cells where they are expressed from specific promoters. Biosensors have extremely high binding specificity because the sensing module is usually obtained by screening lipid-interacting proteins. The fluorescence intensity or hue of a fluorophore or a pair of fluorophores can be modulated by the intrinsic molecular recognition specificity of a biomolecule. Although fluorescent tags are mainly used for localization and expression studies, genetically encoded fluorescent biosensors are increasingly becoming the preferred tools for visualizing and analyzing lipid metabolism in cells, offering high spatial and temporal resolution. Researchers have developed biosensors that recognize specific lipids such as phosphatidylinositol phosphates (PIPs), diacylglycerol (DAG), and phosphatidic acid (PA) (Table 1).

PIPs are a type of phospholipid that contains a phosphorylated inositol head group. Although PIPs make up only a small fraction of total phospholipids, they have important functions in regulatory processes such as intracellular trafficking and cell signaling. To investigate the localization and function of PtdIns3P in living plant cells, yellow fluorescent protein (YFP) was fused to a tandem dimer of the FYVE domain, which specifically binds PtdIns3P, thus creating the genetically encoded fluorescent biosensor YFP-2×FYVE [39]. Another biosensor (YFP-PHPLCδ1) uses YFP to tag PHPLCδ1, the pleckstrin homology domain of human PLCδ1 that specifically binds to PtdIns(4,5)P_2_ [40,41,42]. By expressing YFP-PHFAPP1 in four different plant systems, the turnover of PtdIns4P was shown to occur during various biological processes with different functions [42,43]. Additionally, tagRFP-ML1N*2 probes were engineered by directly fusing fluorescent proteins to the lipid-binding domain of TRPML1, allowing them to quickly respond to changes in PtdIns(3,5)P_2_ levels in living cells [44,45]. Building upon previous research, a new toolbox called the PIPline marker set was developed to study PIPs in Arabidopsis. This toolbox can be used to quantitatively analyze the localizations of various PIPs in relation to known compartment markers [46]. The use of PIP probes has uncovered many new functions of PIPs; for instance, PtdIns3P was shown to affect AHA2-GFP signals at the plasma membrane and induce its internalization [70].

Diacylglycerol kinase (DGK) phosphorylates DAG to generate PA, which serves as a precursor for the biosynthesis of glycerol-phospholipids and triacylglycerol and plays a crucial role as a signaling lipid in regulating various fundamental processes in plants. Therefore, visualizing DAG and PA in the cell is of great significance. The lipid analog BODIPY-PA was employed to study the role of PA produced by phospholipase D in plant tip growth, specifically in elongating tobacco pollen tubes [34]. Live-cell imaging of PA in tobacco pollen tubes was performed using confocal laser scanning microscopy, with the help of a biosensor based on the PA-binding domain of yeast SNARE Spo20p [48]. Furthermore, another biosensor called YFP–C1aPKC was used, which consists of a fusion between YFP and the C1a domain of protein kinase C that specifically binds to DAG [47]. This biosensor allowed DAG to be visualized in living plant cells. Using Förster resonance energy transfer (FRET), a PA-specific optogenetic biosensor called PA-leon was developed to monitor the concentration and dynamics of bioactive PA at the plasma membrane [49]. GFP–N160RbohD, a mobilizable, highly responsive, genetically encoded fluorescent indicator, was recently used to monitor PA dynamics in living cells [50].

In addition to PIPs, DAG, and PA, a few genetically encoded biosensors have been developed to study lipid membrane structure in plants. Membrane dynamics were analyzed using time-lapse fluorescence microscopy and EYFP-RabA4b (a marker for polarized membrane trafficking) in *rhd4-1* mutant plants [51]. TPLATE-GFP (TPLATE, an adaptin-like protein) accumulates along the plasma membrane spanning the division zone. This provided evidence that the plasma membrane undergoes localized endocytosis or membrane remodeling, which are required for the fusion of the cell plate with a predefined region of the plasma membrane [52]. Furthermore, a set of surface charge biosensors was created by fusing an mCITRINE (cYFP) fluorescent protein to a carboxy (C)-terminal farnesyl anchor in conjunction with an adjacent unstructured peptide of varying net positive charges. These biosensors were used to reveal combinatorial lipid codes (such as PtdIns4P, phosphoinositide, PA, and phosphatidylserine) that shape the electrostatic landscape of plant endomembranes [10,30,31].

Special care should be taken when interpreting images obtained using genetically encoded biosensors for lipids in plants. There are two important considerations to keep in mind. First, biosensors are in equilibrium with the free pool of lipids and do not interact with lipids that are bound to effector proteins; therefore, each image represents the instantaneous distribution of the target lipid molecules. Second, the biosensor is expressed in the cell independently of its lipid target; changes in local concentrations of the lipid cause changes in biosensor localization and not overall changes in its expression level or total fluorescence. The use of a FRET biosensor can help mitigate these issues to a considerable extent. To minimize potential off-target interactions, indirect sensors based on DNA/RNA aptamers or optimized binding proteins can be used, as they are less likely to interfere with the native signaling pathway [71,72].

## 5. Current Limitations and New Strategies for Studying Plant Lipids

Fluorophores, such as fluorescent proteins or organic dyes, can be directly attached to antibodies, making them ideal for use in immunoassays. However, immunofluorescence assays are more challenging for lipids than for proteins due to difficulties in experimental operation and a higher risk of artifacts. This is because lipids are poorly immobilized by fixation; moreover, antibodies require a break in the membrane to access internal lipids [73,74]. Another approach is to use fluorophore-labeled lipids, similar to fluorescent protein tags, which have been successfully employed to detect lipid distribution and transport. However, it is important to note that these methods replace a significant portion of the hydrophobic tail or head with a fluorophore. This could interfere with key interactions that drive lipid localization or function, and the resulting fluorescence may not distinguish metabolized lipids from other species [75,76].

To address this issue, our first recommendation is to detect specific lipids with genetically encoded biosensors derived from specific lipid-binding domains. These biosensors have the advantage of being compatible with living cells, enabling the detection of endogenous lipids, and have limited effects on lipid metabolism and distribution when used carefully. Additionally, mass spectrometry imaging (MSI) is an emerging analytical technique for biological samples, offering the visualization of the spatial distribution and relative abundance of lipids [77]. Matrix-assisted laser desorption/ionization mass spectrometry imaging (MALDI-MSI) was successfully used to study the spatial distribution patterns of lipids in cotton (*Gossypium hirsutum*) seeds [78].

Nevertheless, when using fluorescence microscopy, there are several issues that cause artifacts and therefore cannot be ignored. In terms of sensors, there are various factors to consider; for instance, if the binding ratio between lipid molecules and sensor molecules is 1:1, it is necessary to also examine the dissociation coefficient. A 1:1 ratio allows for a direct correlation between the number of bound sensor molecules and the number of lipid molecules, facilitating the quantification of lipids in a sample. The dissociation coefficient is essential for understanding the interaction dynamics between lipids and the sensor, which can significantly influence sensor performance and accuracy. Additionally, it is important to determine whether metabolized lipid molecules will continue to be recognized. Moreover, if a sensor molecule binds to a specific lipid species more tightly compared to its functional interacting molecules, metabolism could be disrupted when the sensor is introduced into a living cell. This can be particularly problematic for analytes present at low levels, although it can be partially remedied by programmed low-level expression.

To achieve binding specificity, it is possible to expand the molecular toolbox by exploring unknown chemical materials in nature. Furthermore, improvements in the detection range, sensitivity to other parameters (e.g., pH and temperature), and intracellular localization can be achieved through molecular simulation, which can provide insights into how molecules interact with each other, as well as through chemical reactions or protein engineering, which allow for adjustments to the structures of individual molecules. Apart from the properties determined by the sensor molecule itself, other factors, such as the signal-to-noise ratio, temporal resolution, spatial resolution, and calibration, also impose constraints on the probe molecule and the detection instrument, often requiring empirical techniques. It is therefore equally important to develop appropriate equipment and standardize experimental technology.

There are indeed several advanced techniques that are being utilized for lipid imaging in plants beyond fluorescence intensity imaging. Generalized polarization (GP) allows for the quantification of membrane lipid order by analyzing the spectral phasor of certain probes (e.g., Di-4-ANEPPDHQ), offering a direct indication of membrane fluidity and lipid packing in plant cells [24,79,80]. An imaging microviscosity toolbox using chemically modified molecular rotors to visualize spatial variations in microviscosity within living plant cells, providing new insights into local mechanobiological processes [81]. These techniques, along with others, provide a comprehensive toolkit for lipid imaging in plants, each offering unique capabilities to probe different aspects of lipid biology, from molecular interactions to supramolecular organization. In Figure 3, we summarize the subcellular localizations of the activities of the various fluorescent molecules discussed in this review. As highlighted in this diagram, the currently available tools for lipid imaging in plants are mainly focused on the cell membrane. New tools must therefore be developed to facilitate the *in vivo* observation of lipid dynamics in organelles. For a more comprehensive understanding of the microscopy methods used for visualizing lipids, readers can refer to a previously published review [82]. Recent advancements in live-imaging techniques, such as variable-angle total internal reflection fluorescence microscopy (VA-TIRFM), structured-illumination microscopy (SIM), and light-sheet fluorescence microscopy (LSFM), continue to evolve rapidly [83]. While these techniques are already suitable for certain research targets (Table 1), it is important to note that the fluorescent molecules for visualizing plant lipids discussed in this review have their own limitations and potential drawbacks. The question remains: How can we effectively overcome these limitations?

### 5.1. New Lipid-Binding Domain for the Design of Biosensors

The light-sensitive domain of the Arabidopsis cryptochrome photoreceptor CY2PHR was used to establish the FRET sensor CY2PHR–CliF. When exposed to light, the molecules form clusters that result in enhanced intermolecular FRET, thereby reporting the intermolecular association of adjacent proteins in living cells [84]. Two-dimensional lipid films resemble cell membranes and provide a suitable method for immobilizing proteinaceous moieties. In addition, intrinsic signal amplification mechanisms have been widely used as biosensing platforms for detecting toxins in environmental and clinical samples [85]. Protein domains can be used to detect the relationship between lipid molecules, and these domains can also be identified in reverse, allowing the detection of proteins or toxins using multifunctional lipid analogs, binding reactions, and digital algorithms [86,87,88]. The proximity-dependent labeling of interacting proteins enables the constant analysis of protein–protein interactions *in vivo* using enzymes to covalently attach reactive groups to nearby proteins [89]. Proximity-dependent labeling can also facilitate the identification of proteins that bind specific lipids.

### 5.2. Novel Strategies Using Fused Fluorescent Molecules

Caffeic acid is a common molecule in plants. A recent study discovered that the fungus Neonothopanus gardneri has a bioluminescence pathway that converts caffeic acid into luciferin involving four enzymes: nnHispS, nnH3H, nnLuz, and nnCPH [90]. In a novel approach, researchers introduced the genes responsible for these enzymes into tobacco plants, allowing reporters to be created that can monitor the activities of plant genes in living tissues over an extended period of time while also investigating their responses to plant hormones [91]. This technology could potentially be used to develop plant biosensors that emit light in response to environmental signals.

Click chemistry is a biocompatible reaction method that involves joining small modular units to efficiently generate new molecules. This technique has proven to be powerful for identifying, locating, and characterizing known and unknown biomolecules by attaching a reporter molecule or substrate of interest to a specific biomolecule. In a study conducted on Arabidopsis tissues, propargylcholine was successfully incorporated into choline phospholipids. The roots readily absorbed propargylcholine, and using click chemistry to attach fluorophores, the subcellular localization of these labeled phospholipids was successfully visualized [92]. TurboID-based proximity labeling has recently emerged as a novel approach to studying the spatial and interaction characteristics of proteins in living plant cells [93]. We anticipate that this method will also be utilized for studying the interactions between lipids and proteins.

## 6. Conclusions

Various tools are available for analyzing plant lipids, although some lipids still cannot be detected. The current tools for visualizing lipid dynamics and homeostasis have some limitations, but when combined and interpreted properly, these techniques can provide valuable insights. The field of genetically encoded biosensors is rapidly advancing, suggesting that real-time imaging techniques will soon enable higher-resolution visualization of the spatiotemporal dynamics and homeostasis of plant lipids. We therefore expect that the current limitations will quickly be overcome, leading to a comprehensive understanding of the molecular mechanisms linking lipid-based signaling and membrane trafficking.

## Figures and Tables

**Figure 1 plants-14-00015-f001:**
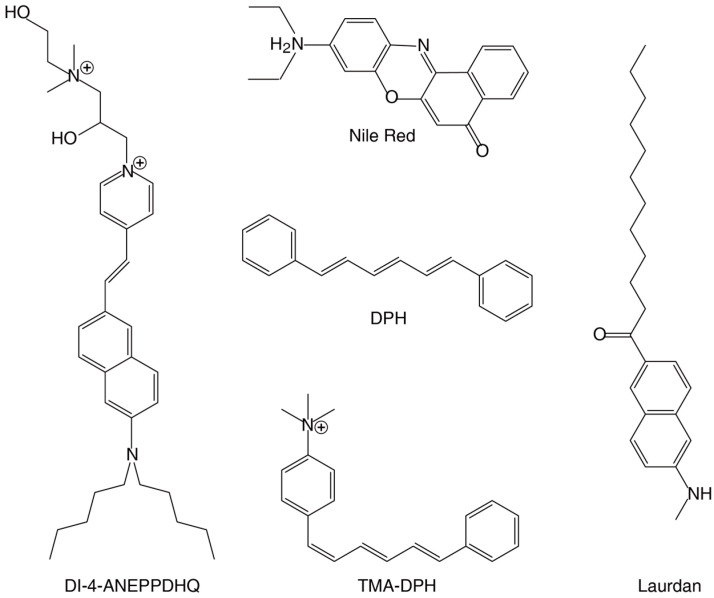
Representative aromatic fluorescent dyes used in plant studies. DI-4-ANEPPDHQ and Laurdan are phase-sensitive membrane probes whose emission spectra exhibit a significant blueshift in the lipid-ordered phase compared with the disordered phase. Nile Red is used to localize and quantitate lipids, particularly neutral lipid droplets within cells. The membrane probes DPH and TMA-DPH are located in the bilayer core and at the interfacial region, respectively. Additional details on the binding specificities of these dyes can be found in Table 1.

**Figure 2 plants-14-00015-f002:**
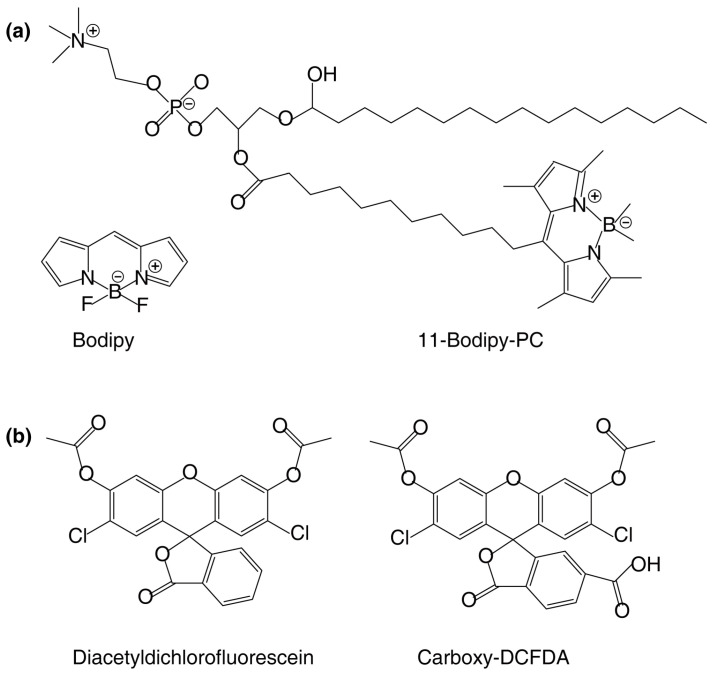
Typical fluorescent lipid analogs utilized in plant studies and their precursors. (**a**) BODIPY-PC incorporated into cells functions as a substrate to generate BODIPY-labelled lipid products. (**b**) Carboxy-DCFDA is colorless and nonfluorescent until the acetate groups are cleaved by intracellular esterases to yield the fluorescent fluorophore. This property makes it a useful tool for visualizing cellular compartments where the intracellular esterases work.

**Figure 3 plants-14-00015-f003:**
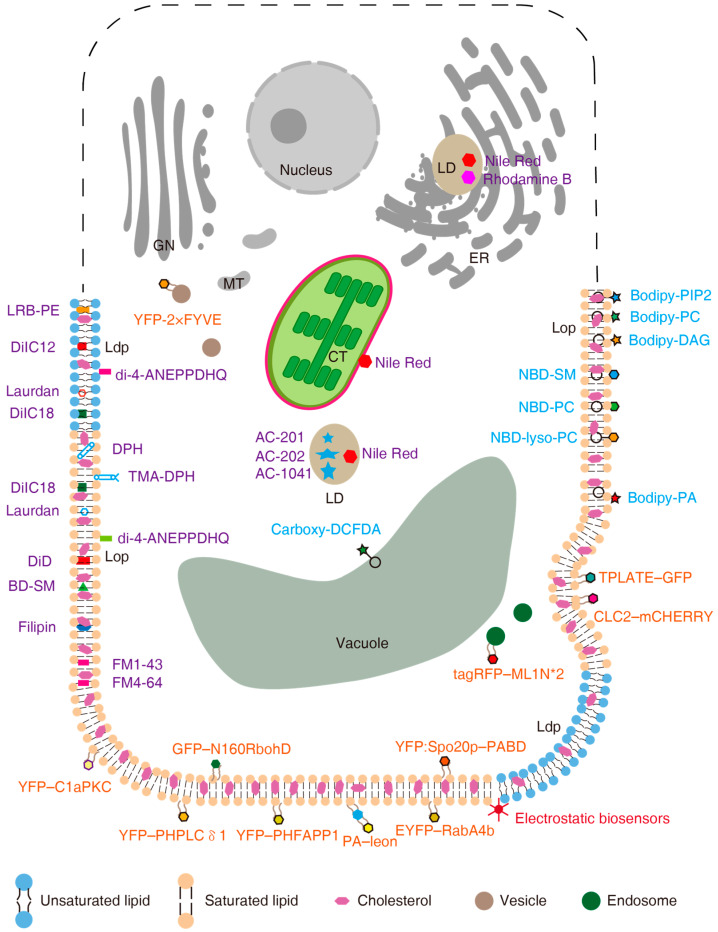
Schematic of the localizations of the activities of various fluorescent molecules. Fluorescent dyes are displayed as simple shapes with labels in purple type. Lipid analogs are represented by colored polygons connected to a circle with blue labels. Genetically encoded biosensors are shown as hexagons with two arms and brown labels. Electrostatic biosensors are fluorescent proteins fused to a carboxy (C)-terminal lipid anchor in conjunction with an adjacent unstructured peptide of varying net positive charge, which makes them able to interact with the membrane (negatively charged by lipids). The major positional information of the probes is shown. GN, Golgi network. MT, mitochondrion. CT, chloroplast. LD, lipid droplet. Lop, liquid-ordered phase. Ldp, liquid-disordered phase.

**Table 1 plants-14-00015-t001:** Fluorescent molecules for visualizing plant lipids mentioned in this review.

Fluorescent Molecules	Target Lipids	Target of Lipids Involved Structure	Research Purposes	Reference(s)
Di-4-ANEPPDHQ, DPH, TMA-DPH, and Laurdan	/	Plasma membrane	The influence of depletion in sterol on the phase behavior of the plasma membrane.	[16]
FM4-64, LRB-PE, DiIC12, DiIC18, DiD, BD-SM, and Laurdan	/	Plasma membrane	To develop lipid staining protocols for plants.	[17]
FM 1-43, FM 4-64	/	Plasma membrane	To test the involvement of lipids turnover in endo- and exocytosis.	[18,19,20,21,22]
Di-4-ANEPPDHQ	/	Plasma membrane	To visualize membrane micro-domains and examine the effect of the change of PM lipid composition on the PM order.	[23,24,25,26]
Ac-201, Ac-202 and Ac-1041	/	Lipid droplets	Characterizing new markers for the investigation of dynamic LD biology within living plant cells.	[27]
Bodipy, Nile Red	/	Lipid droplets	Showing the co-localization of fluorescent proteins and dyes in LD to introduce the method for the protein localization analysis in plants.	[28]
Nile Red	/	Lipid droplets	Investigating lipid bodies formation, localization, trafficking, and plasmodesmata targeting.	[29]
Electrostatic biosensors for PtdIns(4)P, PtdIns(4,5)P2, phosphatidylserine(PS), etc.	/	Membrane lipids with different eletrophilicity	To reveal combinatorial lipid codes shaping the electrostatic landscape of plant endomembranes.	[10,30,31]
Filipin	Sterol	/	Subcellular sterol distribution and intracellular sterol trafficking.	[32,33]
BODIPY–PA	Phosphatidic acid	/	To address the question of whether PA produced by phospholipase D (PLD) participates in plant tip growth in cells, such as elongating pollen tubes.	[34]
BODIPY- tetramethyl-rhodamine-XC6-PtdIns(4,5)P2 [Fluo-PIP2]	PtdIns(4,5)P2	/	To investigate the function of the Arabidopsis thaliana pollen-expressed gene encoding PIP5K4.	[20]
Bodipy-PC	Phosphatidylcholine	/	To assess the impact of Al on the activity of phospholipase.	[35,36]
Carboxy-DCFDA	Phosphatidic acid	/	The potential role of PA signaling in vacuolar dynamics and morphology in the tobacco pollen tube.	[21,37]
NBD-PC and NBD-lysoPC	Phosphatidylcholine	/	To visualize the distribution of themselves and identity their endogenously produced metabolic products.	[22,38]
YFP–2xFYVE	PtdIns3P	/	To investigate localization and function of PtdIns3P in living plant cells.	[39]
YFP-PHPLCδ1	PtdIns(4,5)P2	/	To investigate localization and function of low-abundance PtdIns(4,5)P2 in living plant cells.	[40,41,42]
YFP-PHFAPP1	PtdIns4P	/	To show the turnover of PtdIns4P in multiple biological processes.	[42,43]
tagRFP–ML1N*2	PtdIns(3,5)P2	/	To reliably monitor intracellular dynamics of PI(3,5)P2 in *Arabidopsis* cells.	[44,45]
PIPline marker set	Phosphatidylinositol phosphates	/	To provide a comprehensive collection to study the localization and dynamics of distinct PIP species.	[46]
YFP–C1aPKC	Diacylglycerol	/	To further analyze distribution of diacylglycerol in different plant cells and tissues.	[47]
YFP:Spo20p-PABD	Phosphatidic acid	/	To monitor PA dynamics in plant cells.	[48]
PAleon	Phosphatidic acid	/	To determine the precise spatio-temporal dynamics of PA in living cells and tissues of plants.	[49]
GFP-N160RbohD	Phosphatidic acid	/	Monitoring PA dynamics in living cells.	[50]
EYFP-RabA4b	Phosphatidic acid	/	To analyze membrane dynamics in the process of polarized expansion of root hair cells.	[51]
TPLATE-GFP; CLC2-mCHERRY	Phosphatidic acid	/	To exploring the plasma changing in plant cytokinesis.	[52]

## Data Availability

No new data were created or analyzed in this study. Data sharing is not applicable to this article.

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
