# Peer review of "Imaging Plant Lipids with Fluorescent Reporters"

_plants, 2024, doi:10.3390/plants14010015_

Round 1

Reviewer 1 Report

Comments and Suggestions for Authors

Yong-Kang Li and colleagues present a well written and easy to follow review that encompasses a detailed description of the several approaches currently used in the study of plant lipids by imaging technique. There are only few comments that must be addressed prior to publication.

1)       Line 33: “Membranes can also contain a mix of lipid physical states including gel/solid and liquid disordered or ordered phases.” – Add references. For example: F. Aresta-Branco, A.M. Cordeiro, H.S. Marinho, L. Cyrne, F. Antunes, R.F. de Almeida, Gel domains in the plasma membrane of Saccharomyces cerevisiae: highly ordered, ergosterol-free, and sphingolipid-enriched lipid rafts, J.Biol.Chem., 286 (2011) 5043-5054; and D. Lingwood, K. Simons, Lipid Rafts As a Membrane-Organizing Principle, Science, 327 (2010) 46-50.

2)       Line 71: “These dyes emit specific fluorescence when they come in contact with given metabolites or under certain circumstances.” – Change the sentence to something like “The photophysical properties of these dyes may change according to the surrounding environment or specific molecular interactions.”

3)       In Figure 1, please remove the counter ion (p-Toluenesulfonate) of TMA-DPH. It is not part of the fluorescent probe and may be confusing for the readers.

4)       Line 78: “Sterols and sphingolipids form lipid clusters in the plasma membranes of all cell types in plants, providing a medium for protein signaling complexes and serving as key regions of membrane contact during infection.” – Reference(s) missing.

5)       Line 90 and 95: I believe references 16 and 17 are swapped.

6)       The Authors should take the opportunity and make such a review work more than just a collection of what has been done and with which probe. It is very important that the Authors discuss which probe is more suitable for a specific aim. For example, it is implied that di-4-ANEPPDHQ is a good probe to evaluate membrane order. Although di-4-ANEPPDHQ may be employed to evaluate membrane order, it is not the most suitable probe for that purpose as pointed out in the work by Mariana Amaro, Francesco Reina, Martin Hof, Christian Eggeling and Erdinc Sezgin (Laurdan and Di-4-ANEPPDHQ probe different properties of the membrane, J Phys D Appl Phys. 2017 Apr 5; 50(13): 134004. https://www.ncbi.nlm.nih.gov/pmc/articles/PMC5802044/). Briefly, they state in their conclusions: “…On the other hand, the results for di-4-ANEPPDHQ dye revealed complex relaxation kinetics involving multiple processes. The GPdi-4 values do not correlate with lipid packing and are influenced by cholesterol in a specific way. This discrepancy may result in several factors including interactions between the dye and other membrane components or the exact location of the dye in the membrane [55]. It is of particular importance that di-4-ANEPPDHQ is an electrochromic dye, i.e. its fluorescence emission spectrum is sensitive to the membrane potential. For example, the transmembrane potential of the plasma membrane ranges from around –40 mV to –80 mV, while that of the mitochondrial membrane is around  140 to  180 mV. Consequently, the electrochromic property of di-4-ANEPPDHQ can substantially bias the interpretation of empiric values gathered from cell biology experiments. Therefore, GPdi-4 seems not to be the best indicator of lipid membrane order.” The Authors should discuss this in more detail in their article.

7)       The Authors must specify what each abbreviation stands for when mentioning it for the first time.

8)       It is my understanding that supplementary table 1 should be part of the main text. 

9) In figure 3, please review the relative localization of the probes. For example, DPH and TMA-DPH are at the same level when it is known that DPH is more deeply located in the membrane and TMA-DPH locates closer to the membrane/water interface due to the positively charged trimethylammonium moiety. Di-4-ANEPPDHQ is also located closer to the membrane/water interface due to the positive charges in its structure.

Author Response

Yong-Kang Li and colleagues present a well written and easy to follow review that encompasses a detailed description of the several approaches currently used in the study of plant lipids by imaging technique. There are only few comments that must be addressed prior to publication.

1) Line 33: “Membranes can also contain a mix of lipid physical states including gel/solid and liquid disordered or ordered phases.” – Add references. For example: F. Aresta-Branco, A.M. Cordeiro, H.S. Marinho, L. Cyrne, F. Antunes, R.F. de Almeida, Gel domains in the plasma membrane of Saccharomyces cerevisiae: highly ordered, ergosterol-free, and sphingolipid-enriched lipid rafts, J.Biol.Chem., 286 (2011) 5043-5054; and D. Lingwood, K. Simons, Lipid Rafts As a Membrane-Organizing Principle, Science, 327 (2010) 46-50.

[Response]: We thank the reviewer for this suggestion. We have added these references in the current version (References No.8 and 9)

2) Line 71: “These dyes emit specific fluorescence when they come in contact with given metabolites or under certain circumstances.” – Change the sentence to something like “The photophysical properties of these dyes may change according to the surrounding environment or specific molecular interactions.”

[Response]: We thank the reviewer for the suggestion. We have revised the sentence in the current version on page 2 lines 75-76.

3) In Figure 1, please remove the counter ion (p-Toluenesulfonate) of TMA-DPH. It is not part of the fluorescent probe and may be confusing for the readers.

[Response]: We thank the reviewer for your careful reading. As the reviewer suggested, we have removed the counter ion (p-Toluenesulfonate) of TMA-DPH in Figure 1. We also removed the bromine ion of DI-4-ANEPPDHQ in Figure 1 in the current version.

4) Line 78: “Sterols and sphingolipids form lipid clusters in the plasma membranes of all cell types in plants, providing a medium for protein signaling complexes and serving as key regions of membrane contact during infection.” – Reference(s) missing.

[Response]: Thanks for your comments. We have revised the sentence and added the relevant literatures in the current version (line 85-90 of the tracked version). The details are as follows:

Sterols are isoprenoid derivatives with a characterized planar sterol backbone made up of four condensed aliphatic rings and a hydrocarbon side chain at C17. Higher plants possess a vast array of sterols referred to as phytosterols. Sterols and sphingolipids form lipid clusters in the plasma membranes, providing a medium for protein signaling complexes and serving as key regions of membrane contact during infection [16,17].

16 Bozkurt, T.O., Richardson, A., Dagdas, Y.F., Mongrand, S., Kamoun, S., Raffaele, S. The plant membrane-associated REMORIN1.3 accumulates in discrete perihaustorial domains and enhances susceptibility tophytophthora infestans. Plant Physiol 2014, 165(3), 1005–1018.

17 Ma, Z., Sun, Y., Zhu, X., Yang, L., Chen, X., Miao, Y. Membrane nanodomains modulate formin condensation for actin remodeling in Arabidopsis innate immune responses. Plant Cell 2022, 34(1), 374–394.

5) Line 90 and 95: I believe references 16 and 17 are swapped.

[Response]: We thank the reviewer for your careful reading. We have confirmed the contents of these two references and now they are quoted in right order. (Line 103 and 107 of the tracked version)

6) The Authors should take the opportunity and make such a review work more than just a collection of what has been done and with which probe. It is very important that the Authors discuss which probe is more suitable for a specific aim. For example, it is implied that di-4-ANEPPDHQ is a good probe to evaluate membrane order. Although di-4-ANEPPDHQ may be employed to evaluate membrane order, it is not the most suitable probe for that purpose as pointed out in the work by Mariana Amaro, Francesco Reina, Martin Hof, Christian Eggeling and Erdinc Sezgin (Laurdan and Di-4-ANEPPDHQ probe different properties of the membrane, J Phys D Appl Phys. 2017 Apr 5; 50(13): 134004. https://www.ncbi.nlm.nih.gov/pmc/articles/PMC5802044/). Briefly, they state in their conclusions: “…On the other hand, the results for di-4-ANEPPDHQ dye revealed complex relaxation kinetics involving multiple processes. The GPdi-4 values do not correlate with lipid packing and are influenced by cholesterol in a specific way. This discrepancy may result in several factors including interactions between the dye and other membrane components or the exact location of the dye in the membrane [55]. It is of particular importance that di-4-ANEPPDHQ is an electrochromic dye, i.e. its fluorescence emission spectrum is sensitive to the membrane potential. For example, the transmembrane potential of the plasma membrane ranges from around –40 mV to –80 mV, while that of the mitochondrial membrane is around  −140 to  −180 mV. Consequently, the electrochromic property of di-4-ANEPPDHQ can substantially bias the interpretation of empiric values gathered from cell biology experiments. Therefore, GPdi-4 seems not to be the best indicator of lipid membrane order.” The Authors should discuss this in more detail in their article.

[Response]: We thank the reviewer for your valuable feedback on our manuscript. It is indeed very important to choose appropriate reporters for a specific experimental purpose. To address this, in the revised version, we have included a discussion on the selection of reporters in line 158-169 (tracked version), using Di-4-ANEPPDHQ as an example. Here is the detailed content we have added:

In fluorescence-based lipid imaging, it is imperative to recognize that the efficacy of different reporters can vary significantly when employed for identical experimental objectives. This variation stems from the disparate mechanisms underlying the function of each fluorescent dye. For example, it is implied that Di-4-ANEPPDHQ, frequently used as a polarity sensitive probe, is a good probe to evaluate membrane order, however, it is reported not the most suitable probe for certain circumstance [42]. As Amaro et al (2017) pointed out, since di-4-ANEPPDHQ is an electrochromic dye, the electrochromic property of di-4-ANEPPDHQ can substantially bias the interpretation of empiric values gathered from cell biology experiments. Therefore, it is crucial to discern the mechanisms and relative merits of reporters by referring to specific research papers or by conducting comparative experiments when choosing a reporter for plant lipids studies.

42 Amaro, M., Reina, F., Hof, M., Eggeling, C., Sezgin, E. Laurdan and Di-4-ANEPPDHQ probe different properties of the membrane. J Phys D Appl Phys 2017, 50(13), 134004.

7) The Authors must specify what each abbreviation stands for when mentioning it for the first time.

[Response]: We thank the reviewer for your comments. We have checked through our text to specify what each abbreviation stands for when mentioning it for the first time.

8) It is my understanding that supplementary table 1 should be part of the main text.

[Response]: We thank the reviewer for your comments. We have included the table in the main text now (Line 83 of the tracked version).

9) In figure 3, please review the relative localization of the probes. For example, DPH and TMA-DPH are at the same level when it is known that DPH is more deeply located in the membrane and TMA-DPH locates closer to the membrane/water interface due to the positively charged trimethylammonium moiety. Di-4-ANEPPDHQ is also located closer to the membrane/water interface due to the positive charges in its structure.

[Response]: We appreciate the reviewer's suggestion on the relative localization of the reporters. We have now reviewed the relative localization of certain probes, such as Di-4-ANEPPDHQ, DPH and TMA-DPH. In fact, unfortunately, the majority of the published literatures lack detailed information regarding subcellular localization.

Reviewer 2 Report

Comments and Suggestions for Authors

Li and collaborators provided a reader with a concise review on imaging plant membranes and lipids with fluorescent reporters. The topic is elaborated in sufficient detail to get the flavor of the diversity of experimental approaches while the content is logically structured and coherent. However, there are a few issues listed below that should be addressed by the authors prior to publication their work.

-              Since some of the fluorescent dyes described in the manuscript are not natural lipid analogs and are not used to track individual lipid species but rather lipid membranes and other structures (see e.g. Fig.1), the title should be modified and refer to membrane imaging. Similarly, the title of chapter 2 is also slightly misleading.

-              In the Introduction (e.g. in the third paragraph) alternative microscopic methods for imaging membranes/lipids should be mentioned. For example, Coherent Anti-Stokes Raman Scattering (CARS) Microscopy seem to be appropriate, as the primary strength of this approach lies in the ability of direct imaging lipids (see e.g. DOI: 10.1109/JSTQE.2009.2032512)

-              It is a bit surprising that some more advanced imaging approaches (e.g. generalized polarization (GP), spectral phasor analysis or fluorescence lifetime to study physical properties of membranes are underrepresented in the manuscript These (and some other) issues deserve thorough discussion based on appropriate literature (see e.g. DOI: 10.1016/j.plaphy.2017.08.017; DOI: 10.1016/j.plaphy.2014.12.015; DOI: 10.1186/s42397-020-00054-4; DOI: 10.1073/pnas.192137411).  

Some minor issues include: the phrase “…time-dependent lipid polarization event…” (l.90) requires clarification; it is not true that “Lipid analogs have slight chemical modifications that do not alter their important functional groups…” (l. 146-147); the sentence in l. 227-228 needs rephrasing since YFP-PHFAPP1 and YFP-PHPLCd1 are two different reporters of different specificity; latin names and phrases should be italic.

Author Response

Li and collaborators provided a reader with a concise review on imaging plant membranes and lipids with fluorescent reporters. The topic is elaborated in sufficient detail to get the flavor of the diversity of experimental approaches while the content is logically structured and coherent. However, there are a few issues listed below that should be addressed by the authors prior to publication their work.

1) Since some of the fluorescent dyes described in the manuscript are not natural lipid analogs and are not used to track individual lipid species but rather lipid membranes and other structures (see e.g. Fig.1), the title should be modified and refer to membrane imaging. Similarly, the title of chapter 2 is also slightly misleading.

[Response]: We thank the reviewer for your suggestions. The detailed elucidation that each reporter is used to track individual lipid species, lipid membranes or other related structures are provided in the main text and Table 1.

2) In the Introduction (e.g. in the third paragraph) alternative microscopic methods for imaging membranes/lipids should be mentioned. For example, Coherent Anti-Stokes Raman Scattering (CARS) Microscopy seem to be appropriate, as the primary strength of this approach lies in the ability of direct imaging lipids (see e.g. DOI: 10.1109/JSTQE.2009.2032512)

[Response]: We thank the reviewer for your suggestions. We have supplied more information of alternative microscopic methods (line 45-48 of tracked version). The details are as follows:

Coherent Anti-Stokes Raman Scattering (CARS) microscopy is a non-linear vibrational imaging technique that allows for label-free visualization of cellular lipid storage by harnessing the natural vibrational signatures of lipid molecules, providing high-contrast images of lipid droplets within cells [14].

14 Enejder, A., Brackmann, C., and Svedberg, F. Coherent Anti-Stokes Raman Scattering Microscopy of Cellular Lipid Storage. IEEE J SEL TOP QUANT 2010, 16, 506-515.

3) It is a bit surprising that some more advanced imaging approaches (e.g. generalized polarization (GP), spectral phasor analysis or fluorescence lifetime to study physical properties of membranes are underrepresented in the manuscript These (and some other) issues deserve thorough discussion based on appropriate literature (see e.g. DOI: 10.1016/j.plaphy.2017.08.017; DOI: 10.1016/j.plaphy.2014.12.015; DOI: 10.1186/s42397-020-00054-4; DOI: 10.1073/pnas.192137411).

[Response]: We thank the reviewer for your comments and suggestion. We have added several references and discuss these advanced imaging approaches in the current version (line 368-378 of tracked version). The details are as follows:

There are indeed several advanced techniques that are being utilized for lipid imaging in plants beyond fluorescence intensity imaging. Generalized Polarization (GP) allows for the quantification of membrane lipid order by analyzing the spectral phasor of certain probes (e.g. Di-4-ANEPPDHQ), offering a direct indication of membrane fluidity and lipid packing in plant cells [79, 80, 81]. An imaging microviscosity toolbox using chemically modified molecular rotors to visualize spatial variations in microviscosity within living plant cells, providing new insights into local mechanobiological processes [82]. These techniques, along with others, provide a comprehensive toolkit for lipid imaging in plants, each offering unique capabilities to probe different aspects of lipid biology, from molecular interactions to supramolecular organization.

79 Sena, F., Sotelo-Silveira, M., Astrada, S., Botella, M.A., Malacrida, L., Borsani, O. Spectral phasor analysis reveals altered membrane order and function of root hair  cells in Arabidopsis dry2/sqe1-5 drought hypersensitive mutant. Plant Physiol Biochem 2017, 119, 224-231.

80 Zhao, X., Li, R., Lu, C., Baluska, F.,Wan, Y. Di-4-ANEPPDHQ, a fluorescent probe for the visualisation of membrane microdomains in living Arabidopsis thaliana cells. Plant Physiol Biochem 2015, 87, 53-60.

81 XU, F., SUO, X., LI, F. Bao, C., HE, S., HUANG L., et al. Membrane lipid raft organization during cotton fiber development. J Cotton Res 2020, 3, 13.

82 Michels, L., Gorelova, V., Harnvanichvech, Y., Borst, J.W., Albada, B., Weijers, D., et al. Complete microviscosity maps of living plant cells and tissues with a toolbox of targeting mechanoprobes. Proc. Natl. Acad. Sci. U.S.A. 2020, 117(30), 18110-18118.

4) Some minor issues include: the phrase “…time-dependent lipid polarization event…” (l.90) requires clarification; it is not true that “Lipid analogs have slight chemical modifications that do not alter their important functional groups…” (l. 146-147); the sentence in l. 227-228 needs rephrasing since YFP-PHFAPP1 and YFP-PHPLCd1 are two different reporters of different specificity; latin names and phrases should be italic.

[Response]: We thank the reviewer for your comments. We have addressed these issues. Some of the modified contents are as follows. The latin names and phrases have also been corrected with tracked changes.

Fluorescence recovery after photo-bleaching (FRAP) was employed to measure lipid dynamics, and time-dependent lipid polarization events, where the distribution of lipids, particularly sterols, within the cell is dynamically regulated over time to achieve cellular polarity, were observed to verify the viability of these staining protocols. (line 100-102 of tracked version)

Lipid analogs have slight chemical modifications that do not alter their important functional groups. However, attaching a fluorescent tag to a lipid molecule to create a lipid analog can potentially affect the lipid's function due to several factors, including changes in molecular size, polarity, and interactions with other cellular components [43]. (line 173-176 of tracked version)

Another biosensor (YFP-PHPLCδ1) uses YFP to tag PHPLCδ1, the pleckstrin homology domain of human PLCδ1 that specifically binds to PtdIns(4,5)P2. (line 255 of tracked version)

43 Sezgin, E., Can, F.B., Schneider, F., Clausen, M.P., Galiani, S., Stanly, T.A., et al. A comparative study on fluorescent cholesterol analogs as versatile cellular reporters. J Lipid Res 2016, 57(2), 299-309.
